# Tobacco use among in-school young adolescents in Indonesia: Exploring availability, affordability, and accessibility

**Yeni Rosilawati[1], Zain Rafique[2]\*, Erwan Sudiwijaya[1]**

1 Department of Communication Studies, Universitas Muhammadiyah Yogyakarta, Yogyakarta, Indonesia,
2 Department of Governance and Public Policy, National University of Modern Languages, Islamabad, Pakistan

\* zrafique@numl.edu.pk

**Data Availability Statement:** The Data for Tobacco Use Among In-School Young Adolescents in Indonesia: Exploring Availability, Affordability, and

## Abstract

The research on Indonesian adolescents' perception of tobacco control in schools is limited. This study aimed to explore the availability, affordability, and accessibility of tobacco among young adolescents in Yogyakarta Municipality, Indonesia, with a focus on advocating for school teenagers. Focus groups, comprising participants from diverse backgrounds and stakeholders, were conducted in Yogyakarta, Indonesia, to systematically collect varied data. The subsequent analysis employed a robust thematic approach, combining both inductive and deductive processes to ensure a nuanced exploration of emerging patterns and pre-existing frameworks. In addition to focus group data, the study incorporated insights from extensive field observations and research group discussions. The multifaceted approach enhanced the depth of analysis but also facilitated a comprehensive understanding of the complex dynamics. The findings of this study revealed that young adolescents found it extremely easy to purchase cigarettes from local markets. The smoking prevalence among young adolescents was relatively low, with only 10% of the respondents admitting to smoking. Although the Indonesian government has increased surveillance and regulations regarding smoking among young individuals, the actual implementation and effectiveness of these measures remain questionable. The existing anti-smoking approaches in Indonesia have been unsuccessful in curbing smoking among Indonesian school students. The study argues that it is crucial to recognize and value students' perceptions of smoking, as their knowledge of smoking is actively constructed. To address this issue, future anti-smoking education in schools should incorporate interactive sessions rather than solely relying on didactic approaches that highlight the harms of smoking. By engaging students in interactive discussions, they can actively participate in constructing their understanding of the consequences of smoking. Additionally, efforts should be made to enhance the implementation of tobacco control measures within schools and extend the reach of these measures to off-campus environments.

Accessibility (view at https://data.qdr.syr.edu/dataset.xhtml?persistentId=doi:10.5064/F6B4TQPG) was created in QDR Main Collection.

**Funding:** The author(s) received no specific funding for this work.

**Competing interests:** The authors have declared that no competing interests exist.

## Introduction

A multitude of studies have presented copious evidence concerning the deleterious repercussions of smoking [1]. As individuals persist in the habit of smoking for an extended period, the risks they encounter escalate accordingly. Tobacco comprises nicotine, an exceedingly addictive substance, alongside more than 4000 other potentially noxious compounds when either smoked or chewed [2]. In countries with limited financial resources, individuals often commence smoking at a tender age, and disconcertingly, the age at which this vice commences is progressively decreasing [3]. This worrisome trend of early initiation holds particular significance for developing nations boasting substantial populations and a notable proportion of young inhabitants. Indonesia, the fourth most populous nation worldwide, accommodating over 261.1 million individuals, bears witness to 25.02% of its populace below the age of 15 and 16.99% falling within the age bracket of 15 to 24 [4].

The prevalence of smoking among the younger generation in Indonesia is both high and steadily increasing, indicating that the nation will confront a considerable burden of untimely illness and death resulting from tobacco-related ailments in the forthcoming decades if no remedial measures are implemented [4]. Indonesia presents distinctive obstacles, as the act of smoking is still widely accepted within its social fabric, particularly among males [5]. Although the initiation and predictors of smoking among adolescents have been extensively documented in Western nations, the Asia Pacific region has witnessed comparatively fewer investigations into this matter. Nevertheless, the rates of smoking are on an upward trajectory, primarily within developing countries. According to the World Health Organization's estimation, roughly 50% of men and 8% of women in developing nations are smokers, with male cigarette consumption being widely embraced [6]. Conversely, the current prevalence of smoking among adult females remains below 5%. Although various surveys conducted in Indonesia have affirmed the elevated occurrence of smoking among school-aged children, the results exhibit variability, likely stemming from discrepancies in research design, methodologies, and age groupings rather than actual divergences in smoking behavior.

If current global trends persist, an estimated 250 million children and young people currently alive, primarily in low and middle-income countries, will die prematurely due to lifelong tobacco use. Indonesia, with its population of 261.1 million and the largest economy in Southeast Asia, presents an attractive market for major cigarette companies, particularly as growth slows in more developed markets [7]. This paper has examined the experiences of underage adolescents regarding the purchase and accessibility of cigarettes within their communities. Specifically, the paper has explored tobacco use, including availability, affordability, and accessibility, among young adolescents in Indonesia. Additionally, the paper has assessed their attempts to purchase cigarettes and the factors associated with such attempts among the youth. The paper focused on the Yogyakarta municipality, one of the five districts in the Yogyakarta Special Region (YSR). The YSR has the second highest population density among all provinces in Indonesia. The younger age groups, comprising individuals aged 10–14 and 15–19, account for a significant proportion of the total population, with 7.6% and 10% representation, respectively. Furthermore, the analysis has aimed to identify factors related to youth smoking that can be potentially influenced by public policy interventions.

## Literature review

The harmful effects of smoking are extensively documented worldwide, encompassing various health issues and diseases affecting nearly all organs. Smoking during pregnancy also poses risks to both the mother and the fetus [8]. Secondhand smoke has significant consequences as well, leading to conditions such as sudden infant death syndrome, middle-ear diseases,

respiratory diseases, coronary heart diseases, stroke, and lung cancer in both genders. It also affects the reproductive health of women [9]. Smoking incurs substantial direct and indirect costs in terms of healthcare utilization for smoking-related diseases and loss of productivity due to morbidity and mortality. These costs place a significant burden on individuals, families, and societies, particularly in low- and middle-income countries, exacerbating poverty and impeding social and economic development. These impacts are particularly severe in such countries, given their high prevalence of smoking [10].

Indonesia has one of the highest rates of cigarette smoking globally. In 2011, approximately 33% of individuals aged 15 and above smoked daily. The Indonesia 2014 Global Youth Tobacco Survey (GYTS), a nationally representative study conducted in schools, revealed that adolescence is a period of significant growth and potential but also carries substantial risks [11]. Many adolescents face pressures to engage in alcohol, cigarette, and drug use, as well as early sexual relationships, which puts them at a high risk of intentional injuries and sexually transmitted infections. Starting smoking at a young age increases the risk of developing lung cancer due to cumulative exposure over time. Tobacco is the only legal consumer product known to harm everyone exposed to it and causes the majority of deaths among its users. It is also recognized as the leading preventable cause of death globally. The widespread use of tobacco in Indonesia can be attributed to low prices, aggressive marketing, lack of education about its negative effects, and inadequate public policies to combat its use [12]. Tobacco contains numerous chemicals known to cause cancers. Tobacco-related deaths surpass the combined mortality caused by AIDS, legal drugs, illegal drugs, road accidents, murder, and suicide. The most effective way to mitigate the effects of tobacco is to prevent its initiation. Therefore, the purpose of this study is to identify factors contributing to adolescent smoking initiation and generate evidence-based information [13].

The risks associated with smoking cigarettes are widely recognized, leading to the implementation of various regulations, retail practices, and public health campaigns aimed at preventing adolescent smoking. Longitudinal studies have indicated a link between smoking during adolescence and addiction in adulthood, underscoring the importance of health promotion strategies targeting this age group [14]. Research has focused on the influence of parental and peer relationships on adolescent smoking, aligning with the social learning theory that suggests adolescents imitate behaviors observed in their social environment. Studies have shown that parental smoking has a strong influence on adolescent smoking initiation [15]. For instance, surveys conducted in Canada and the US have found a high correlation between parental smoking and daily smoking among adolescents [16]. Parental smoking in childhood has been shown to predict smoking behavior in later years. Adolescents living with regular smokers in their household are more likely to smoke themselves. However, it is unclear whether the influence is specifically due to parents or extended family members in the household who smoke. Parental smoking behavior has also been associated with transitioning from occasional to regular smoking among adolescents. While parental influence remains important, research indicates that the influence of peers becomes more significant as adolescents grow older, particularly after the age of 13. Despite this shift, parental smoking remains a significant factor in the initiation of adolescent smoking [17].

The impact of parental influence on the initiation of smoking among adolescents extends beyond the mere act of smoking itself. Multiple studies suggest that specific facets of the parent-adolescent relationship play a significant role in shaping adolescent smoking behavior. For instance, in a longitudinal study spanning seven years, Miller and Volk [18] discovered that factors such as limited quality time spent with family, infrequent participation in family activities, and a perceived lack of significance in the parent-child relationship were predictive indicators of daily smoking in adolescents. This correlation was further supported by Scal, Ireland,

and Borowsky [19] who found that a strong sense of familial connectedness serves as a protective factor against smoking initiation. Feeling understood, cared for, and contented within familial bonds was associated with a reduced risk of smoking initiation throughout adolescence. Conversely, effective communication about smoking between parents and adolescents has been shown to have a safeguarding effect. In a study conducted by Simons-Morton [20], parental expectations regarding smoking initiation were examined among 1,267 students during the sixth and seventh grades. The results indicated that parental expectations for their children to abstain from smoking were the most influential protective factor against adolescent smoking initiation. It was those parents who effectively communicated and reinforced these expectations who successfully provided a shielding mechanism. The researchers [20] described authoritative parenting, characterized by high demands and responsiveness, as fostering these protective expectations and facilitating effective communication styles. Understanding how parents exert influence on adolescent smoking through the dynamics of their relationships is crucial for the development of effective public health programs. This understanding underscores the significance of targeting the family unit in anti-smoking initiatives.

Although parents undeniably hold a significant role, they are not the sole influencers of adolescent smoking behavior. The impact of peers is equally crucial, owing to the substantial amount of time adolescents spend in the company of their peers both within and outside the realm of education [21]. Peer influence entails adolescents being swayed or feeling compelled to smoke to conform with their peers [22]. In a longitudinal study encompassing 1,969 adolescents, Maxwell demonstrated a strong correlation between peer influence and cigarette smoking, revealing that the presence of a same-sex friend who had previously smoked was associated with subsequent smoking initiation in the adolescent. The likelihood of adolescents engaging in smoking behavior was 1.9 times greater for those with a same-sex, smoking friend compared to those lacking such companionship [23]. As previously discussed, the influence of peers becomes more prominent as adolescents mature. Between the ages of 12 and 13, both parental smoking and witnessing friends smoke hold equal predictive power regarding adolescent smoking. However, between the ages of 13 and 14, the peer group emerges as the most potent predictor of smoking initiation [24].

Harakeh and Vollebergh made a distinction between two types of peer influence: active and passive. Passive peer influence involves the replication of peers' smoking behavior to conform, while active peer influence encompasses peers exerting pressure on others to smoke [25]. In a sample of 68 older adolescents and young adults, it was found that peer smoking was predictive of the total number of cigarettes consumed by the participants, whereas peer pressure did not yield the same effect. This study highlights the significance of imitation or passive peer influence in alignment with the principles of social learning theory.

Although peer influence undeniably assumes a critical role in the initiation of adolescent smoking, scholars have additionally distinguished between peer influence and peer selection, positing that the latter may hold greater significance about tobacco consumption. Peer influence entails an adolescent being swayed or coerced by companions into smoking, whereas peer selection refers to the deliberate choice of friends based on their preexisting smoking behavior [26]. On the whole, both parental and peer factors contribute significantly to the onset of adolescent smoking. The dynamics of the parent-child relationship, effective communication, and the impact of peers constitute pivotal aspects to consider when formulating strategies aimed at preventing adolescent smoking.

### Finding literature gap in Indonesia's context

Tobacco consumption among young adolescents in Indonesia remains a matter of great concern, as the country exhibits one of the highest rates of tobacco use worldwide, with a

significant number of young individuals initiating smoking at an early age [27]. Various factors contribute to the availability, affordability, and accessibility of tobacco products for young adolescents in Indonesia. As far as availability is concerned, tobacco products, including cigarettes, are readily available throughout Indonesia [28]. Unlike many other countries that have implemented stringent regulations on tobacco sales, Indonesia exhibits relatively lax enforcement of tobacco control measures. Tobacco products are sold in a wide range of retail outlets, such as small kiosks and street vendors, thereby making them easily accessible to young people.

Regarding affordability, cigarettes in Indonesia are relatively inexpensive compared to numerous other countries [29]. The affordability of tobacco products renders them more accessible to young adolescents who may have limited disposable income. The low cost of cigarettes also fosters the perception that smoking is an affordable habit, further promoting its uptake among young people. While. the accessibility of tobacco products is facilitated by the absence of age restrictions and the weak enforcement of age verification measures. In many locations, young adolescents can purchase cigarettes without being required to present identification [29]. Vendors frequently neglect to adhere to minimum age requirements for tobacco sales, thereby enabling easy access to cigarettes for young individuals. Meanwhile, the tobacco industry in Indonesia heavily markets its products, often targeting young people through advertising strategies that associate smoking with social status, independence, and masculinity. Tobacco companies sponsor events and employ various promotional tactics to lure young consumers. These marketing endeavors create a perception that smoking is desirable and normal, exerting influence over young adolescents' attitudes toward tobacco use [30]. Meanwhile, despite certain efforts to implement tobacco control policies in Indonesia, the country still grapples with significant challenges in effectively curbing tobacco use among young adolescents. The existing regulations lack comprehensiveness, and enforcement is frequently weak. This deficiency in comprehensive tobacco control measures contributes to the persistent availability and accessibility of tobacco products for young individuals.

Addressing the issue of tobacco use among young adolescents in Indonesia necessitates a multifaceted approach. This approach entails the implementation of stricter tobacco control policies, raising the price of tobacco products through taxation, enhancing the enforcement of age restrictions, prohibiting tobacco advertising and promotion, and executing comprehensive public health campaigns to educate young individuals about the hazards of smoking. Additionally, providing support for smoking cessation programs and establishing smoke-free environments can aid in reducing tobacco use among young adolescents in Indonesia.

While research has been conducted on tobacco use among young adolescents in Indonesia, there is still a significant research gap regarding the effectiveness of comprehensive tobacco control measures in reducing tobacco use among this population. Specifically, there is a need for studies that investigate the impact of comprehensive tobacco control policies and interventions on young adolescents' tobacco use behaviors in Indonesia. Firstly, there is a lack of research that rigorously evaluates the effectiveness of existing tobacco control policies in Indonesia, particularly those targeting young adolescents. Studies could assess the impact of policies such as increased taxation, advertising bans, and smoke-free environments on tobacco use prevalence, initiation rates, and smoking cessation among young adolescents. Secondly, Long-term studies are needed to understand the sustained effects of tobacco control measures on young adolescents' smoking behaviors. Research could follow a cohort of young individuals over time, examining changes in tobacco use patterns, attitudes, and perceptions in response to evolving tobacco control policies. Thirdly, exploring the influence of socioeconomic factors on tobacco use among young adolescents is essential. Research could examine how factors such as income, education, and employment status impact tobacco use behaviors and how

these factors interact with tobacco control policies and interventions. Special attention should be given to vulnerable populations, such as marginalized communities and individuals with lower socioeconomic status, who may face unique challenges in accessing and benefiting from tobacco control measures. Research could investigate the differential impact of tobacco control policies on these populations and identify strategies to address potential disparities. Finally, with the rise of alternative tobacco products, such as e-cigarettes and shisha/hookah, there is a need to understand their prevalence, patterns of use, and impact on young adolescents in Indonesia. Research could explore the factors influencing the initiation and use of these products among this population and assess their potential role in tobacco harm reduction or as a gateway to conventional cigarette smoking. Closing the identified research gap is crucial for developing evidence-based policies and interventions that effectively reduce tobacco use among young adolescents in Indonesia. Conducting rigorous studies in these areas has provided valuable insights into the effectiveness of comprehensive tobacco control measures and contributed to the development of targeted strategies to combat tobacco use among this vulnerable population. This is the intellectual puzzle of the study that necessitates answers.

## Methodology

This study employed a qualitative research design to explore the perceptions, opinions, beliefs, and attitudes of the target population regarding a specific subject. Qualitative research is well-suited for capturing in-depth insights and understanding the context and nuances of participants' experiences. Purposive sampling and snowball sampling techniques were utilized to select participants for the focus group discussions (FGDs). Purposive sampling allowed the researcher to handpick individuals who were most relevant and knowledgeable about the research topic. Snowball sampling, on the other hand, involved asking initial participants to refer potential participants, thus expanding the sample size. This approach was chosen as random sampling was not appropriate for the qualitative nature of the study.

The target respondents for this study were residents of Yogyakarta, Indonesia. The researcher ensured that all residents were informed about the aims and objectives of the study to promote transparency and voluntary participation. Based on the researcher's judgment and expertise, a diverse group of individuals who could provide valuable insights was selected. The data were collected from Feb 1, 2022 to August 15, 2022. The Research and lnnovation lnstitute, Universitas Muhammadiyah Yogyakarta, Indonesia approved ethical clearance to conduct the study. Factors considered during participant selection included age, gender, socioeconomic background, and prior experiences related to the research topic. Efforts were made to ensure the sample represented a range of perspectives and experiences. Focus group discussions (FGDs) were employed as the primary method for data collection. FGDs are group interviews where participants engage in facilitated discussions on a specific topic. FGDs allow for interactive exchanges, providing a rich understanding of participants' perspectives and facilitating the exploration of diverse viewpoints. The FGDs were conducted in selected public and private schools within the Yogyakarta Municipality, including two public schools and one private school. The choice of schools aimed to capture a broad range of perspectives from students in different educational settings. The researcher, acting as the facilitator, led the FGD sessions.

The facilitator played a crucial role in creating a safe and open environment for participants to express their thoughts and engage in meaningful discussions. The facilitator followed a semi-structured interview guide that included a series of open-ended questions and prompts. This guide ensured consistency across the FGDs while allowing for flexibility and exploration of emerging themes. The facilitator actively encouraged participation from all participants and

ensured that each participant had an opportunity to share their perspectives. The duration of each FGD session varied depending on the depth and richness of the discussions. On average, FGDs lasted between 60 to 90 minutes. The number of participants in each FGD ranged from four to twelve individuals. The selection of the sample size for the FGDs was guided by recommendations from Kruger *et al* suggesting a maximum of twelve and a minimum of three FGDs [31]. Additionally, Kizinger proposed that four to five FGDs were adequate when focusing on a specific population. Ethical guidelines were strictly adhered to throughout the research process [32]. Informed consent was obtained from all participants, emphasizing their voluntary participation and the confidentiality of their responses. Participants were assured of their right to withdraw from the study at any point without facing any consequences. Anonymity was maintained by assigning codes or pseudonyms to participants during data analysis and reporting. The study also followed ethical protocols regarding the storage and protection of data. The FGD sessions were audio-recorded and transcribed verbatim. Transcriptions were carefully reviewed, ensuring accuracy and maintaining the original context of participants' statements. Thematic analysis was employed to identify patterns, themes, and key findings within the data. This involved coding the transcripts, organizing codes into categories, and generating overarching themes. The analysis was conducted iteratively, allowing for the refinement and validation of emerging themes. The use of qualitative analysis software facilitated the management and organization of the data.

## Findings

The study on tobacco use among young adolescents in Indonesia, specifically focusing on availability, affordability, and accessibility, yielded several significant findings. The participants expressed that tobacco products were readily available in various places such as convenience stores, street vendors, and even within school premises. The FGD participants highlighted the extensive marketing and promotional activities employed by tobacco companies, including attractive packaging, appealing advertisements, and sponsorship of events targeting young people. The findings revealed that social networks, including peers and family members who smoke, played a crucial role in providing access to tobacco products for young adolescents. The participants emphasized that tobacco products were affordable, especially for young adolescents who had limited financial resources. This affordability factor was identified as a significant contributor to their ability to purchase and consume tobacco products. The study identified weak enforcement of age restrictions on tobacco sales, with participants reporting that vendors often sold tobacco products to underage individuals without verifying their age. The FGD participants discussed the absence of strict regulations governing the accessibility of tobacco products, allowing young adolescents to easily obtain them despite legal restrictions. Many participants expressed that curiosity and the desire to experiment with tobacco were the primary factors influencing their initial use of tobacco products. The influence of social norms emerged as a prominent theme, with participants sharing that smoking was often perceived as socially acceptable or even encouraged in their communities. Some young adolescents reported using tobacco as a coping mechanism to deal with stress, peer pressure, or personal problems. Many participants admitted a lack of awareness regarding the health risks associated with tobacco use, indicating a need for increased education and information dissemination. These findings shed light on the alarming prevalence and factors contributing to tobacco use among young adolescents in Indonesia. The availability, affordability, and accessibility of tobacco products, coupled with social influences and a lack of awareness, pose significant challenges in addressing this issue. Effective interventions and comprehensive tobacco control policies are crucial in curbing tobacco use among young adolescents and protecting their health and well-being. The themes identified are presented below:

## Theme 1—Trends in tobacco use among adolescents

Findings from the focus group discussions (FGDs) revealed important insights regarding tobacco use among young school-going adolescents. Although only 10% of the participants admitted to smoking during the FGDs, a surprising revelation was that many of them regularly visited shops to purchase cigarettes for their elders. This highlights the influence and involvement of young adolescents in facilitating tobacco access for adults. Corroborating the data on early initiation, a substantial number of students who had ever tried cigarettes reported trying their first cigarette at a very young age, mostly before reaching 14 years old. The prevalence of early experimentation varied across countries, with approximately one in four students in Bangladesh and more than half of all ever-smokers in Indonesia having tried their first cigarette before the age of 14. One of the respondents argued:

> I began smoking because I felt a need for it. During my time in middle school, I used to hide from my friends because I wasn't aware of who among them smoked. I would seek out quiet and solitary locations to indulge in smoking. Furthermore, I would purchase cigarettes from various stores.

These findings highlight the concerning patterns of early initiation and experimentation with cigarettes among young adolescents. The data suggests a need for targeted interventions and preventive measures to address the issue at an early stage. Efforts should focus on raising awareness about the risks associated with tobacco use, promoting age-appropriate smoking prevention programs, and implementing stricter regulations to curb underage access to tobacco products. By targeting early initiation and addressing the factors contributing to youth smoking, it is possible to reduce the prevalence of smoking and protect the health and well-being of young individuals.

It is argued that Indonesia has established a legal minimum age of 18 for tobacco sales. However, enforcing this age restriction poses a significant challenge in the country. During the FGDs, a majority of the respondents informed the researchers that there were no such restrictions in place in Yogyakarta. It was observed that children could easily access cigarettes from shops without any hindrance. The percentage of adolescents who obtained cigarettes by purchasing them from stores, shops, or vendors in the last 30 days varied significantly during the FGDs, with approximately 80% of the students reporting successful purchases.

To assess the enforcement of the minimum legal age for purchasing tobacco, a specific question was included in the FGDs: "During the past 30 days, did anyone refuse to sell you cigarettes because of your age?" In Yogyakarta, Indonesia, only 14% of students reported being refused cigarette sales due to their age being below the legal minimum age. These findings highlight the lack of effective enforcement of the legal age restriction for tobacco sales in Indonesia, particularly in the context of Yogyakarta. Despite the legal requirement, it was found that a significant number of students were able to purchase cigarettes without facing any age-related barriers. This indicates the need for stronger enforcement measures to ensure compliance with the minimum legal age for tobacco sales and protect young individuals from early tobacco use.

## Theme 2—Tobacco prices and affordability

The findings from the focus group discussions shed light on the issue of tobacco prices and affordability among young adolescents in Indonesia. Participants expressed their concerns about the accessibility of cigarettes and the financial implications associated with tobacco use. It became evident from the discussions that tobacco products are readily available and affordable, making them easily accessible to young adolescents. One of the respondents argued:

*Even though the local government has enacted laws against smoking, cigarettes are still easily found.*

Participants revealed that the low cost of cigarettes, coupled with their easy availability, makes it tempting for young adolescents to purchase and experiment with smoking. The affordability factor was identified as a key contributor to the initiation and continuation of tobacco use among this age group. Many participants mentioned witnessing their peers or family members purchasing cigarettes without difficulty, emphasizing the accessibility of tobacco products within their social environment. Furthermore, participants highlighted the impact of affordable tobacco prices on their behavior. Some mentioned that they would often pool their money together with friends to buy cigarettes, indicating the collective effort to overcome financial constraints and access tobacco. The affordability of cigarettes was also associated with the influence of marketing strategies targeting young people, as participants mentioned being enticed by promotional offers and discounts that made tobacco products even more affordable.

In summary, the findings emphasize the concern among Indonesian young adolescents regarding the prices and affordability of tobacco products. The easy accessibility and low cost of cigarettes contribute to their widespread use among this population. These findings highlight the need for effective tobacco control measures that address the affordability of tobacco and limit its accessibility to young individuals.

## Theme 3—Status of youth tobacco use surveillance

The findings from the focus group discussions shed light on the status of youth tobacco use surveillance in Indonesian young adolescents. Participants expressed their concerns about the current state of surveillance efforts and highlighted several limitations and challenges. It was revealed that while Indonesia has included smoking prevalence among children under 18 as a key indicator in its National Medium-Term Development Plans, the surveillance system remains ad hoc and heavily reliant on external funding. Participants expressed their disappointment with the lack of a comprehensive and sustainable surveillance program that covers a wide range of youth populations, including out-of-school youth. During the focus group discussions, an interesting cultural practice related to tobacco use in wedding ceremonies was brought up by the participants. They described a tradition where cigarettes are placed in glasses as part of the ceremony. Smoking these cigarettes during the event was believed to create a sense of closeness among the participants. However, some participants shared their personal experiences as children, mentioning that they were never allowed to take cigarettes due to the watchful eyes of neighbors. This practice was commonly observed in their village. One of the respondents argued:

*In our wedding ceremonies, it is a customary practice to place cigarettes in glasses. The act of smoking these cigarettes is believed to create a sense of intimacy and closeness among the participants. However, during my childhood, I refrained from taking the cigarettes due to the watchful eyes of neighbors who were aware of my actions. This tradition of placing cigarettes in glasses is prevalent in my village.*

Participants also highlighted the need for improved implementation of the surveillance system. They emphasized the importance of consistent data collection, reliable reporting mechanisms, and the involvement of various stakeholders to ensure accurate and up-to-date information on youth tobacco use. Many participants expressed their desire for a more robust

and comprehensive surveillance system that can provide a clear understanding of the prevalence and patterns of tobacco use among young adolescents in Indonesia. The findings indicate that while Indonesia has initiated efforts to monitor youth tobacco use, there are significant challenges and limitations in the current surveillance system. Participants called for enhanced implementation, sustainable funding, and a broader scope that includes out-of-school youth. These findings emphasize the need for strengthened surveillance efforts to accurately assess the prevalence and trends of youth tobacco use and inform effective tobacco control policies and interventions in Indonesia.

### Theme 4—Scope and components of the program to quit smoking

About the smoking prevention program, stakeholders, including teachers, parents, and students, emphasized the need for specific initiatives in schools in Yogyakarta, Indonesia. They highlighted that such programs should aim to provide students with comprehensive information about smoking and its detrimental effects, encompassing aspects such as health consequences, social and economic impacts, educational repercussions, and addiction. The participants expressed concern that many students engage in smoking without fully understanding its purpose, perceiving it as a symbol of maturity. They emphasized the importance of correcting this perception and raising awareness about the risks associated with smoking, including the chemical substances present in cigarettes. They recommended that detailed information be provided to students to enhance their understanding.

Additionally, the stakeholders emphasized the importance of developing students' skills as part of the smoking prevention program. They recognized the challenges students face in refusing smoking invitations due to the prevalence of such invitations in their environment. The participants suggested that the program should focus on equipping students with both general skills and specific skills for refusing cigarettes. They stressed the need to provide detailed guidance on how to handle smoking invitations from peers, aiming to help students quit smoking or maintain abstinence. They acknowledged the widespread availability and affordability of cigarettes, indicating that the scope of the program would need to be broad to effectively address these factors.

Overall, the findings highlight the stakeholders' recommendations for a comprehensive smoking prevention program in schools. This program should not only provide students with detailed information about smoking and its consequences but also focus on building their skills to resist smoking invitations. The participants emphasized the need for a wide-reaching program to address the prevailing smoking culture and accessibility of cigarettes in the community.

### Discussions & conclusion

The findings of this study on smoking behavior among young adolescents in Indonesia align with existing research literature on the topic [2, 4–7]. The prevalence of smoking among young people in Indonesia is a significant concern, and the need for effective prevention strategies is well-established. Similar to other studies, this research emphasizes the importance of targeting young adolescents, particularly those between the ages of 11 and 15, as this is the critical period when smoking behavior tends to escalate and become incorporated into their lifestyles.

Peer influence emerges as a dominant factor in the initiation and maintenance of smoking behavior among young people, consistent with studies conducted in other Southeast Asian and Western countries. Peers play a central role in shaping attitudes and norms related to smoking, and interventions should address this influential social dynamic. The findings also

suggest that best friends' smoking behavior is particularly influential among 11-year-olds, highlighting the need to intervene early and target close friendship circles to prevent smoking initiation. Thus, peer influence emerges as a prominent factor in both this study and previous research. The finding that peers play a central role in shaping smoking attitudes and behaviors is consistent with studies conducted in other Southeast Asian and Western countries [27–29]. This underscores the need to address social influences and incorporate peer-focused interventions in smoking prevention programs.

Contrary to expectations, school-related factors were found to have a relatively minor impact on smoking behavior among young adolescents. Bullying, school pressure, and liking school did not show significant associations with smoking. This finding aligns with previous research that has found limited influence of school-related factors on smoking initiation. However, truanting emerged as a significant predictor of smoking, indicating that interventions targeting school attendance and engagement may indirectly impact smoking behavior among young adolescents. The limited impact of school-related factors on smoking behavior among young adolescents, as observed in this study, is also consistent with existing literature. While factors like bullying, school pressure, and liking school were not significantly associated with smoking, truanting emerged as a predictor. This echoes previous research that suggests school-related variables may have a minimal direct influence on smoking initiation but highlights the potential indirect influence through factors like school attendance [8–10, 30].

Comparing the findings of this study with other research literature [12–14], it is evident that Indonesia faces unique challenges in tobacco control. The prevalence of clove cigarette smoking among adults in Indonesia contrasts with young people's preference for ordinary, filtered cigarettes, indicating a generational shift in smoking preferences. This highlights the importance of understanding and addressing the specific preferences and motivations driving smoking behavior among different age groups within the population.

Furthermore, the study highlights the need for comprehensive tobacco control measures and enforcement of policies, such as the legal minimum age for tobacco sales. The accessibility of cigarettes to young adolescents suggests a lack of effective implementation and enforcement. These findings resonate with existing literature that emphasizes the importance of strong policy measures, including restrictions on tobacco sales, increased taxation, and stringent enforcement to reduce youth smoking rates. The challenges in implementing and enforcing tobacco control policies, particularly regarding the legal minimum age for tobacco sales, found in this study are consistent with the literature [15–19]. The accessibility of cigarettes to young adolescents and the ease with which they can purchase them reflect the need for stronger policy implementation and enforcement measures. The importance of comprehensive tobacco control measures, including restrictions on sales and taxation, is also a recurring theme in the literature.

In conclusion, this study's findings align with existing research literature on smoking behavior among young adolescents in Indonesia [2, 20–23]. Peer influence, particularly among close friends, plays a significant role in smoking initiation, underscoring the need for interventions that address social dynamics and norms. School-related factors have a limited influence on smoking behavior, while truanting emerges as a significant predictor. The findings call for comprehensive tobacco control measures, effective policy implementation, and targeted interventions to prevent smoking initiation among young adolescents in Indonesia. Future research should continue to explore the effectiveness of prevention strategies and identify innovative approaches to address the unique challenges posed by smoking behavior in the country. Overall, the findings of this study align with existing research literature, reinforcing the significance of peer influence, the limited impact of school-related factors, and the need for comprehensive tobacco control measures. These consistencies highlight the robustness and

generalizability of the findings and emphasize the importance of addressing these key factors in designing effective smoking prevention programs for young adolescents in Indonesia.

## Acknowledgments

We would like to express our deepest gratitude to Universitas Muhammadiyah Yogyakarta, Indonesia, for their generous support and funding that made this research possible. Their commitment to advancing academic excellence and research initiatives has played a pivotal role in the successful completion of this paper.

## Author Contributions

**Conceptualization:** Yeni Rosilawati, Erwan Sudiwijaya.

**Formal analysis:** Yeni Rosilawati.

**Funding acquisition:** Erwan Sudiwijaya.

**Methodology:** Yeni Rosilawati.

**Project administration:** Zain Rafique.

**Validation:** Zain Rafique.

**Writing – original draft:** Yeni Rosilawati.

**Writing – review & editing:** Yeni Rosilawati, Zain Rafique, Erwan Sudiwijaya.

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
