## [Decision Letter · Decision Letter 0]

31 Aug 2023

PONE-D-23-17148Tobacco Use Among In-School Young Adolescents in Indonesia: Exploring Availability, Affordability, and AccessibilityPLOS ONE

Dear Dr. Rafique,

Thank you for submitting your manuscript to PLOS ONE. After careful consideration, we feel that it has merit but does not fully meet PLOS ONE’s publication criteria as it currently stands. Therefore, we invite you to submit a revised version of the manuscript that addresses the points raised during the review process.

Please note that we have only been able to secure a single reviewer to assess your manuscript. We are issuing a decision on your manuscript at this point to prevent further delays in the evaluation of your manuscript. Please be aware that the editor who handles your revised manuscript might find it necessary to invite additional reviewers to assess this work once the revised manuscript is submitted. However, we will aim to proceed on the basis of this single review if possible. 

We look forward to receiving your revised manuscript.

Kind regards,

Jianhong Zhou

Staff Editor

PLOS ONE

Journal Requirements:

Reviewers' comments:

Reviewer's Responses to Questions

**Comments to the Author**

1. Is the manuscript technically sound, and do the data support the conclusions?

Reviewer #1: Yes

2. Has the statistical analysis been performed appropriately and rigorously? 

Reviewer #1: Yes

3. Have the authors made all data underlying the findings in their manuscript fully available?

Reviewer #1: No

4. Is the manuscript presented in an intelligible fashion and written in standard English?

Reviewer #1: No

5. Review Comments to the Author

Reviewer #1: The authors of the manuscript should not have been identified in the copy sent to the reviewer

The abstract section has a methods component that is too brief and a conclusion section that is unnecessarily long. The authors should rectify this.

The manuscript is full of grammatical and typographical errors in all segments. The authors should pay attention to this ie:

Even when the local government have been enacted law against smoking, the cigarettes is still easy to be found

The objectives and the research problem section in the introduction was written in future tense rather than past tense even though the research has been conducted.

The introduction is unnecessarily long and the justification for the manuscript did not focus on the research problem. Rather, a lot of unnecessary concepts were highlighted.

6. PLOS authors have the option to publish the peer review history of their article (what does this mean?). If published, this will include your full peer review and any attached files.

Reviewer #1: No

---

## [Author Response · Author response to Decision Letter 0]

2 Feb 2024

Dear Mr. Zhou,

I appreciate the swift evaluation of my manuscript and the constructive feedback provided by the reviewer. I am grateful for the opportunity to revise and resubmit the manuscript to address the points raised during the review process. I have carefully considered the comments and suggestions from the reviewer, and I am confident that the revised version will significantly enhance the quality and clarity of the manuscript. I would like to express my gratitude to the reviewer for their valuable insights.

In response to the reviewer's comments, I have prepared a detailed rebuttal letter addressing each point raised during the review. This letter is attached as a separate file labeled 'Response to Reviewers.' Additionally, I have made the necessary revisions to the manuscript, which are highlighted in the marked-up copy provided as a separate file labeled 'Revised Manuscript with Track Changes.' Finally, the unmarked version of the revised manuscript is included as a separate file labeled 'Manuscript.'

I have adhered to the provided deadline and submitted all required documents through the Editorial Manager platform. Should there be any additional information required or if further clarification is needed, please do not hesitate to contact me. I am committed to working collaboratively to ensure that the revised manuscript meets the standards of PLOS ONE.

Thank you for your time and consideration. I look forward to the opportunity to have my work re-evaluated.

Best regards,

Zain Rafique

REVIEWER 1 COMMENT AUTHOR RESPONSE PAGE NUMBER

The authors of the manuscript should not have been identified in the copy sent to the reviewer

 Apologies for the oversight. In the revised manuscript, I've redacted author information to comply with the double-blind review process. The corrected 'Revised Manuscript with Track Changes' is now submitted. 

The abstract section has a methods component that is too brief and a conclusion section that is unnecessarily long. The authors should rectify this. The methods section in the abstract has been expanded for greater clarity, while the conclusion has been appropriately condensed. I believe these adjustments have significantly improved the manuscript, addressing the concerns raised during the review process. The corrections are highlighted in red color. Please refer to page 2

The manuscript is full of grammatical and typographical errors in all segments. The authors should pay attention to this ie:

Even when the local government has enacted law against smoking, cigarettes is still easy to be found Thank you for your thorough review and valuable feedback. I acknowledge the presence of grammatical and typographical errors in the manuscript and appreciate your highlighting this concern. In the revised version, I have diligently addressed these issues, conducting comprehensive proofreading and editing to ensure the clarity and correctness of the language throughout the manuscript.

I trust that these improvements will enhance the overall quality of the manuscript, and I look forward to your reconsideration. For instance, I have changed suggested line to Even though the local government has enacted laws against smoking, cigarettes are still easily found.

The objectives and the research problem section in the introduction was written in future tense rather than past tense even though the research has been conducted.

 I acknowledge the oversight in the introduction section, specifically in the objectives and research problem, where I inadvertently used future tense instead of past tense despite the completion of the research. I have rectified this issue in the revised manuscript, ensuring that the introduction accurately reflects the completed nature of the study. The corrections are highlighted in the red color. Please refer to page 3 & 4

The introduction is unnecessarily long and the justification for the manuscript did not focus on the research problem. Rather, a lot of unnecessary concepts were highlighted. I have carefully considered your comments regarding the length of the introduction and the need for a more focused justification related to the research problem. In the revised manuscript, I have significantly shortened the introduction, ensuring conciseness, and restructured the justification to emphasize the core research problem while eliminating unnecessary concepts.

I believe these changes enhance the manuscript's clarity and alignment with the research focus. I look forward to your evaluation of the revised version. Please refer to first 3 4 pages

---

## [Decision Letter · Decision Letter 1]

14 Mar 2024

Tobacco Use Among In-School Young Adolescents in Indonesia: Exploring Availability, Affordability, and Accessibility

PONE-D-23-17148R1

Dear Dr. Rafique,

We’re pleased to inform you that your manuscript has been judged scientifically suitable for publication and will be formally accepted for publication once it meets all outstanding technical requirements.

Kind regards,

Yogesh Kumar Jain, MPH

Academic Editor

PLOS ONE

Additional Editor Comments (optional):

Reviewers' comments:

Reviewer's Responses to Questions

**Comments to the Author**

1. If the authors have adequately addressed your comments raised in a previous round of review and you feel that this manuscript is now acceptable for publication, you may indicate that here to bypass the “Comments to the Author” section, enter your conflict of interest statement in the “Confidential to Editor” section, and submit your "Accept" recommendation.

Reviewer #1: All comments have been addressed

2. Is the manuscript technically sound, and do the data support the conclusions?

Reviewer #1: Yes

3. Has the statistical analysis been performed appropriately and rigorously? 

Reviewer #1: N/A

4. Have the authors made all data underlying the findings in their manuscript fully available?

Reviewer #1: No

5. Is the manuscript presented in an intelligible fashion and written in standard English?

Reviewer #1: Yes

6. Review Comments to the Author

Reviewer #1: There has been a significant improvement in the manuscript and the issues raised have been addressed.

7. PLOS authors have the option to publish the peer review history of their article (what does this mean?). If published, this will include your full peer review and any attached files.

Reviewer #1: **Yes: **Dr Oyapero Afolabi

---

## [Editor Report · Acceptance letter]

18 Mar 2024

PONE-D-23-17148R1 

PLOS ONE

Dear Dr. Rafique, 

I'm pleased to inform you that your manuscript has been deemed suitable for publication in PLOS ONE. Congratulations! Your manuscript is now being handed over to our production team.

Kind regards, 

on behalf of

Dr. Yogesh Kumar Jain 

Academic Editor

PLOS ONE